# Software Correction of Speed Measurement Determined by Phone GNSS Modules in Applications for Runners

**DOI:** 10.3390/s23052678

**Published:** 2023-03-01

**Authors:** Pawel Biernacki

**Affiliations:** Department of Acoustics, Multimedia and Signal Processing Faculty of Electronics, Fotonics and Microsystems Wroclaw University of Science and Technology, 50-350 Wroclaw, Poland; pawel.biernacki@pwr.edu.pl; Tel.: +48-713-203-035

**Keywords:** digital filters, estimation, GNSS accuracy, speed measurement

## Abstract

This paper presents the results of a study on software correction of speed measurements taken by GNSS receivers installed in cell phones and sports watches. Digital low-pass filters were used to compensate for fluctuations in measured speed and distance. Real data obtained from popular running applications for cell phones and smartwatches were used for simulations. Various measurement situations were analyzed, such as running at a constant speed or interval running. Taking a very high accuracy GNSS receiver as the reference equipment, the solution proposed in the article reduces the measurement error of the traveled distance by 70%. In the case of measuring speed in interval running, the error could be reduced by up to 80%. The low-cost implementation allows simple GNSS receivers to approach the quality of distance and speed estimation of very precise and expensive solutions.

## 1. Introduction

The latest approach to increasing personal activity, managing weight and improving health is through technological advances, activity trackers and smartphones. These approaches can be effective due to reduced costs, a user-friendly environment [1,2], real-time information and feedback [3,4].

In today’s world, running has become a very popular form of physical activity. A lot of people around the world practice this sport. By wanting to follow their achievements and progress, runners reach for running applications on their smartphones. The most popular are Runkeeper, Strava, Adidas miCoach and Nike Run. They use the phone’s built-in GNSS modules to measure the users’ routes. The operation of most GNSS modules is based on measuring the location at a frequency of 1Hz and from that using the proper software designating the route, speed and acceleration. Running apps use this data to determine average speed, maximum or minimum speed over a given distance, distance traveled, difference in elevation over the route and also calories burned. For the runner, knowledge of the actual values of these parameters is important to assess current training and plan further training or competitions.

GNSS accuracy is affected by a number of factors that generate noise in GNSS data. The accuracy of GNSS measurements can be improved by setting up GNSS signal relay stations (differential GPS); in practice, data accuracy problems remain and setting up relay stations is expensive and time-consuming [5].

To make sense of GNSS satellite signals, the prediction problem must be solved [6]. How can a set of observed variables (GNSS signals) predict unobserved variables (the movement of an athlete). One important first step to solving the prediction problem is to filter out the stochastic noise in the GNSS stream that results from the measurement inaccuracies of the GNSS system. For such real-time processing, where only part of the dataset is available, extrapolation is required [7].

One way to filter GNSS data is to use various curve fitting techniques such as moving average smoothing, digital smoothing, polynomial filter and local regression smoothing. The smoothed GNSS data can then be defined as a less noisy runner movement path [8]. The runner moves at variable speeds and has a lot of freedom in terms of space, time and direction. Simple smoothing techniques in this case can lead to the loss of valuable data, such as the point where curving or backward motion begins and ends. Kalman filtering methods are one option among the available options for straightening real-time GNSS data. Kalman filters are commonly used to map moving objects based on collected position or real-time data [9,10]. However, they require additional hardware and are not cheap to implement. Augmented methods require additional equipment or 3rd party help to establish the position, some examples are:The 3-GNSS based system, which is formed by three antennas forming an equilateral triangle [11].The combination of differential GPS and inertial measurement units (IMU) [12].WAAS-enabled GPS (system with ground reference stations) and IMU [13].

Many works [14,15,16] focus on the static positioning accuracy problem for cheap receivers. They bypass the issue of the accuracy of determining the position in the case of movement. The aim of this paper is to present a GNSS data measurement correction made in post processing to achieve the better accuracy of outdoor running and walking velocity for cheap commercial sport equipment. The proposed solution uses simple logic to establish the runner activity state and two IIR filters to correct the speed and acceleration calculated by the GNSS receiver. The effectiveness of the proposed method for amateur GNSS devices (sports watches and smartphones) has been verified by comparing the achieved results with professional GNSS signal recorders.

## 2. Methods

A single healthy and physically active participant took part in all data collection tests. The workouts were divided into two types. The first was a continuous run of 5, 10 or 20 km. The second was an interval run of 100, 200 or 500 m performed in intervals of 100 or 200 m. Each type of run was recorded on the same route. A single workout was one test record. The route for testing is shown in Figure 1. It includes several 90-degree turns and two tunnels under the railroad, where GNSS signal loss was notable.

The same protocol was repeated on 50 occasions (running sessions). Each time, the GNSS receivers were working at a frequency of 1Hz (one measurement per second). The running sessions were preceded by 3 days of rest, so that the runner’s fatigue would not affect their running pace. Before each test, the GNSS equipment was upgraded with the newest ephemeris data.

The participant wore the Garmin Forerunner 255 (GPS, Glonass, Galileo) and Polar Pacer (GPS, Glonass) watches on the left wrist and a smartphone (Samsung S10, Xiaomi Redmi 9A) with the Runkeeper and Strava applications activated on the left and right arms. A Polar South X6 with support for GPS (L1/L2), GLONASS (L1/L2), Galileo (E1C/E5A/E5B and Beidou (B1/B2/B3) with 10cm accuracy was used as the reference GNSS receiver. The receiver was attached to the runner’s belt. The distance and the speed were compared against an objectively measured distance and velocity with the reference equipment.

## 3. The Problem of Accuracy of Position Measurement in GNSS

To determine position, a GNSS receiver must measure the distance to at least four satellites. A computer built into the receiver calculates the position, altitude and speed based on microwave radio signals transmitted at the speed of light (where the satellite is the receiver). The main sources of GNSS errors are [17,18]:Ephemera errors;The inaccuracy of the time pattern;Multi-path of satellite signals;Antenna phase center variations;Receiver noise;Ionospheric delays;Tropospheric delays;Geometric errors in the alignment of satellites relative to the receiver.

These errors can generate inaccuracies in determining the position of the receiver of about 10 m or more (Figure 2).

The runner themself can contribute to the mismeasurement of their position by [19]:Fast turning or turning around (Figure 3);Rapid braking or acceleration;Covering the satellite with their body (signal loss).

Adding up the aforementioned and other variables (relativistic errors [20], for example, are not listed), the potential errors in measuring the GNSS receiver’s position can make the application determine the running parameters inconsistently and mislead its user.

## 4. Assumptions of the Running Parameter Estimation Software System

The proposed running parameter estimation system is based on digital averaging (low-pass) filters. Their task is to level the errors described in Section 2 by reducing the fluctuations (variance) of the measured quantities around the true values. It was assumed that in the running applications, that function in a programmatic way, it is not possible to react to individual components of the error (runner independent errors described above). The exception here is the number of satellites currently available. It is possible to skip the measurements that were made for the number of satellites below a certain threshold.

Instead, the system should be able to respond to the user’s running pattern. Cases such as acceleration, running intervals or rapid changes in direction of movement must be taken into account in the estimation of running parameters. This requires the use of intelligent recognition and processing. It becomes necessary here to use, in addition to the GNSS module, an accelerometer, which is installed in modern cell phones.

Particularly critical are the moments of sudden changes in running speed. Their detection guarantees a correct indication of speed and distance traveled. A situation disliked by runners is the need to stop (lights at pedestrian crossings, a passing vehicle in their path, etc.). Time is counted and the distance covered does not change. The average speed and the speed for a given distance decrease, which is frustrating when analyzing such results, especially when the ’fight’ is for every single second. In such a situation, the system should stop operation.

## 5. Selected Elements of the Running Parameter Estimation Software System

A schematic diagram of the running parameter estimation system is shown in Figure 4.

Data from the accelerometer and GNSS module control the entire ’intelligence’ of the system. Based on this, the current state of the runner’s location is identified. Three possibilities are distinguished:‘Stop’: The runner stands. No updates are made to the running parameters and the time is stopped. Averaging filter parameters are set for the ’change of pace’ state.‘Change of pace’: The runner changes running speed. The averaging filter parameters are changed to account for faster changes in the runner’s speed (averaging at a lower level, i.e., a mild low-pass filter).‘Steady run’: The runner maintains a constant speed. Filters average the received data from the GNSS module as much as possible (sharp low-pass filter).

The run parameter estimation block consists of two low-pass filters with infinite impulse responses (IIR) [21,22] placed in cascade (Figure 5).

Firstly, this solution gives better averaging results compared to a single filter. Secondly, the IIR filters used are of order 2, which means that they require very little computing power for their operation. There is no need to use finite impulse response filters here, since their linear phase characteristics in the pass band are not needed, and they are much more computationally demanding than IIR filters with the same amplitude characteristics.

In addition, based on the data from the GNSS module (NMEA frames), it is known how many satellites are in the field of view of the phone. When this number is insufficient, then the given position measurement is ignored.

## 6. Results

In order to test the performance of the running parameter estimation system, a test application was designed and implemented in Android (minimum version 4.3) for the Samsung and Xiaomi phones used for testing. Data were collected simultaneously from the test app and from Adidas’ miCoach running program. Since it was not possible to intervene in the software of sports watches, their tests were carried out offline after collecting GNSS data on the computer. The MATLAB 2021a package was used to quickly determine IIR averaging filter coefficients.

Sample test results are shown below. Figure 6 shows the performance of the speed estimation block for a steady run (a run with a constant heart rate that does not tire the runner too much). The fluctuation in the GNSS module reading can be clearly seen. The variance of the results is at the level of 0.2 km/h, which for an hour-long run at 12 km/h results in a difference of 1 min from the assumed time at the finish line (that is significant). You can also see the ‘loss’ of satellites (sharp ‘pins’ around the 7th and 20th kilometers). The estimation block (IIR filters) worked correctly. Defective measurements were missed (the threshold of satellites needed was assumed to be five—theoretically four is enough, but when they are in an unfavorable position with respect to the GNSS receiver the measurements can be subject to large error). The determined trend of running speed is at a constant level (the slight lowering is due to the fatigue of the runner).

Figure 7 and Figure 8 show the more difficult to estimate case of interval (variable) running. Accelerometer data were used to determine the runner’s acceleration/deceleration moments. A certain threshold of tempo change was assumed, which determines the detection of the starting point of the tempo change, here 3 km/h. This is the most sensitive parameter of the whole system. The estimation of the moment of acceleration/deceleration controls the switching on/off of the estimation block, i.e., the results generated by the whole system. The tests show that most of the false rate changes were due to loss of satellite visibility and were a short-term effect. Nevertheless, it is rather impossible to eliminate all such situations. It should also be kept in mind that with a GNSS operating frequency of 1 Hz, we only have 60 measurements per minute at our disposal. For filters with high inertia (small forgetting constant), returning from a false rate change to the real one can take a long time and greatly influence the final result.

Figure 9 shows the effect of the choice of filter coefficients on the result of running speed estimation. It can be seen that the use of two IIR filters gives much better results (red and green colors on the graph) in estimating the speed trend than a single one (blue color). The tests have shown that a single set of coefficients will not give good results for highly variable and uniform gears. The use of a single filter is a compromise here. We obtain worse results for a constant pace, but a faster response to changes in pace. Of course, swapping the set of IIR filter coefficients during operation is possible, but requires additional logic (decision rules).

Figure 10 and Figure 11 show turning around in the same place without and with the working proposed system. All runs took the same route. The run in (Figure 11) was undertaken to compare turning around in different place.

To compare the results from the reference equipment, the following two measures were proposed:(1)$$Error_{distance}=\frac{|distance_{measured}-distance_{reference}|}{distance_{reference}}*100\%$$
for distance measurement, and
(2)$$Error_{speed}=\frac{\sum_{t=0}^{T}\left|speed_{measured}\left(t\right)-speed_{reference}\left(t\right)\right|}{\sum_{t=0}^{T}speed_{reference}\left(t\right)}*100\%$$
for speed measurement, where *T* is the measurement time interval.

Table 1 shows the mean value of measured distance accuracy for Garmin and Polar watches, Samsung and Xiaomi smartphones and the proposed solution tested on watch data. The smartphones give the biggest error. For a distance of 10 km the error is as much as 103 m. If one runs with a pace of 5 min/km, this results in a 30 s difference from the correct result. After correction, the error is 33 m, which gives a 10 s difference.

The results after the speed corrections are shown in Table 2. Speed “10 × 100 m × 100 m” refers to 10 × 10 m, with a pace of 4 min/km, and a 100 m break with a pace of 6 min/km. Again, the smartphones results are the worst. For a running speed of 5 min/km, an error value of 1% means a difference of 3 s in covering one kilometer. For short distances this is not a big error, but for long distances, such as a half-marathons or marathons, this results in minutes of difference. For runners competing for lifetime records, this can mean a wasted season of preparation. The proposed solution makes it possible to reduce the error by half for a strongly fluctuating run pace and by 80% in the case of a more constant one.

## 7. Discussion

All measurements were taken on the same road. They differed in length and duration. The results shown in the tables were averaged for a given training unit. It was assumed that the accuracy of the test devices was at a similar level (as suggested by the catalog data). Running paces may have varied from one training unit to the next, but the differences were not large (a few seconds per minute).

The proposed system is intended to be an overlay for built-in processing systems in phones or watches. The selection of appropriate values of filter coefficients here is a key element of correct operation of the system. They depend mainly on the intensity of the run. Tests have shown that for continuous running (one pace) the filters should be more low-pass. For interval runs, they should take into account rapid changes in the runner’s pace. The ideal solution would be to change the filter coefficients depending on the type of the runner’s workout. The system should learn the way the user runs. It seems that the cadence of steps here could be an element that differentiates the type of training. Changing the cadence means changing the pace. Such a parameter could control the characteristics of the filters, making them more or less low-pass.

Another problem is the loss of signal (such as running through a tunnel). It has been noticed that this results in a drop in running speed, and only when the device ‘sees’ the satellite do the results return to the correct values. The proposed solution uses data from the accelerometer to determine whether the runner is moving. If so, the last measurement from the visible satellite is used as an input signal to the filters. Figure 12 shows such a situation.

Running through the tunnel took about 7 s. The smartphone lost the signal and decreased the estimated speed by almost 1 min per kilometer for a distance of 250 m. In this case, the proposed system took the cadence (number of steps per minute calculated by reading the accelerometer data) to detect the runner’s movement and excite the filters with data from before the signal was lost.

The filtered distance estimate can be used to determine a runner’s route. In some cases, it was noted that the corrected distance value used for calculating the runner’s position at a particular time moment could improve their localization on the route map. The result for a steady run is shown in Figure 13. The real route is on the bike road (yellow line in figure). The line denoted by the number 3 was obtained from the Xiaomi phone and the Runkeeper application. The red line below, which runs exactly on the bicycle path, was determined based on data after distance correction. 

Such a solution did not always give good results. It was possible to obtain more precise (consistent with the map) results after offline processing, when the direction of the runner was known and it was easier to determine which position coordinates of the runner to correct.

## 8. Conclusions

The results obtained from the test studies of the gear parameter estimation system allow the following conclusions:Commercial phone systems are based on raw GNSS measurements and give results with an error that is too large for aspiring runners;Popular sports watches are not suitable to measure fast speed changes with high accuracy;IIR filters, which are switched on based on runner activity and satellite availability, can improve the estimation of running parameters by actually determining the runner’s speed. The performance improvement can reach 80% in relation to popular solutions for runners available on the market;Greater improvements in the accuracy of distance and running speed estimations were obtained for phones. It seems that they have a worse GNSS receiver than those in watches. It is then easier to achieve better estimation improvement;The choice of filter coefficients affects the estimate of the running speed;A critical to good estimation is the determination of spikes in running pace.

The proposed solution seems to be ideal for a workout course known in advance (based on training plans stored in the phone). Otherwise, the runner should learn to run with the phone or sports watch to be able to select the parameters of the system individually for each workout. Building a universal system (for any type of workout) is a challenge that encourages further research and testing.

## Figures and Tables

**Figure 1 sensors-23-02678-f001:**
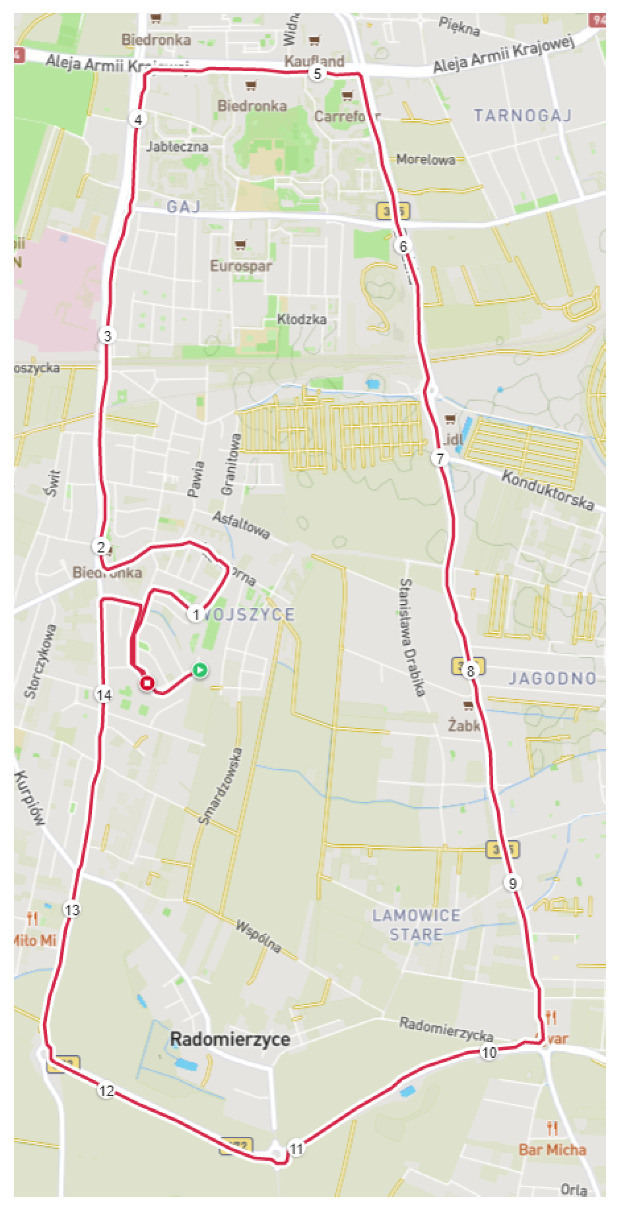
Route used for testing (numbers denote consecutive kilometers on the route).

**Figure 2 sensors-23-02678-f002:**
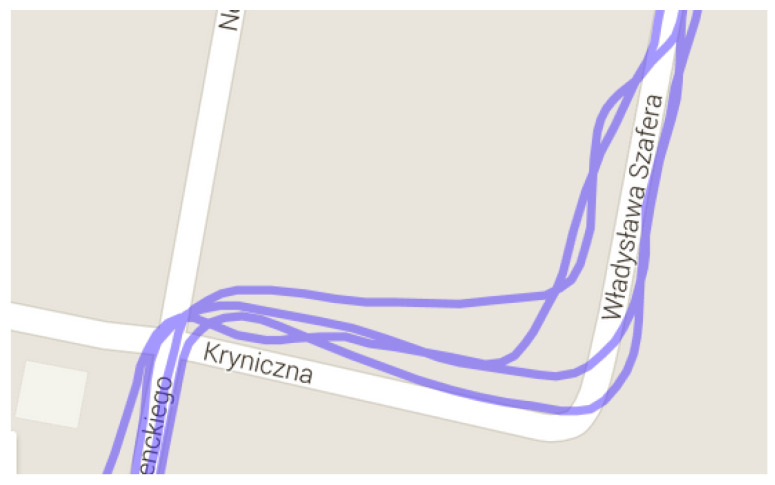
GNSS receiver inaccuracy–running the same route (Polar watch data).

**Figure 3 sensors-23-02678-f003:**
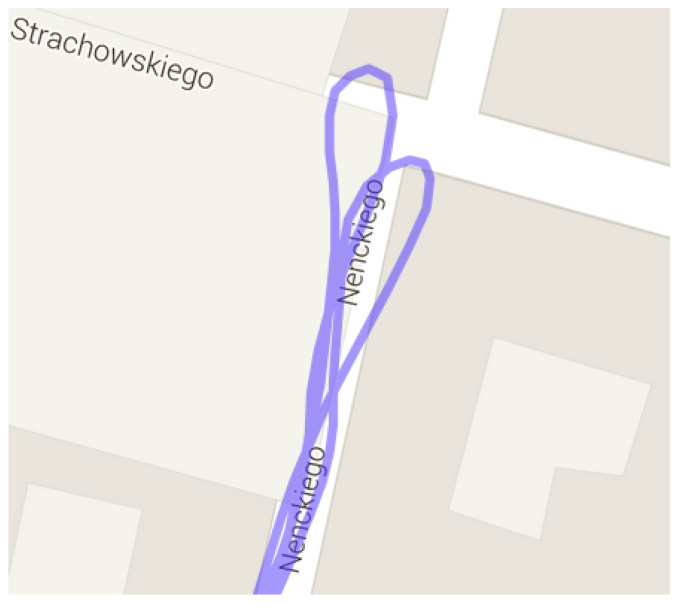
GNSS receiver inaccuracy: turning around in the same place (Garmin watch data).

**Figure 4 sensors-23-02678-f004:**
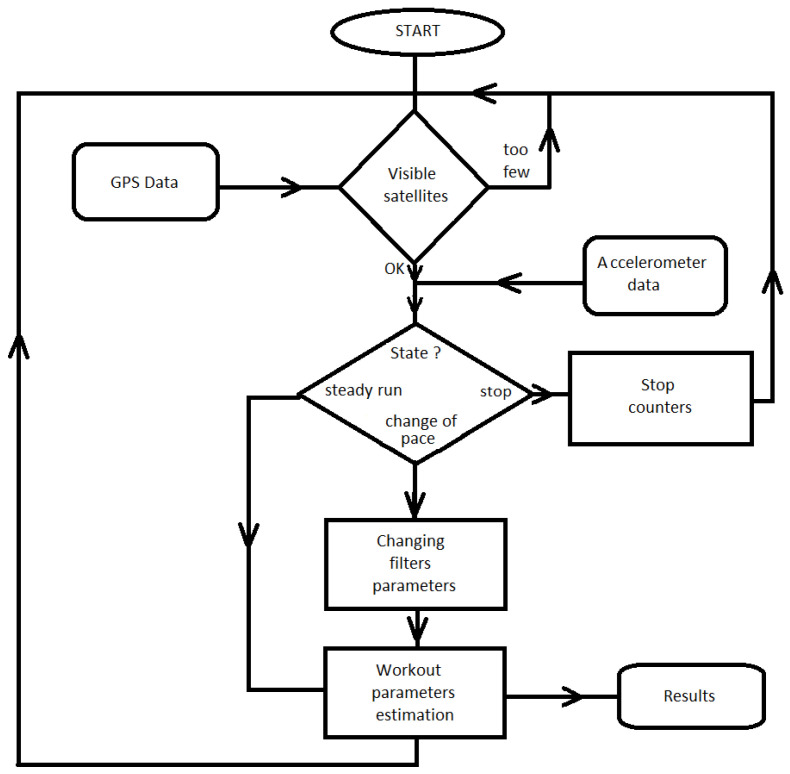
Functional diagram of the system for estimation of running parameters.

**Figure 5 sensors-23-02678-f005:**
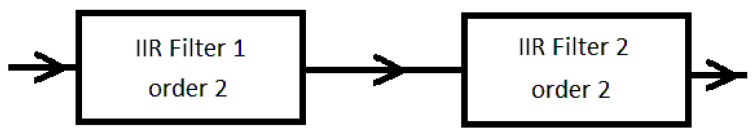
Run parameter estimation block.

**Figure 6 sensors-23-02678-f006:**
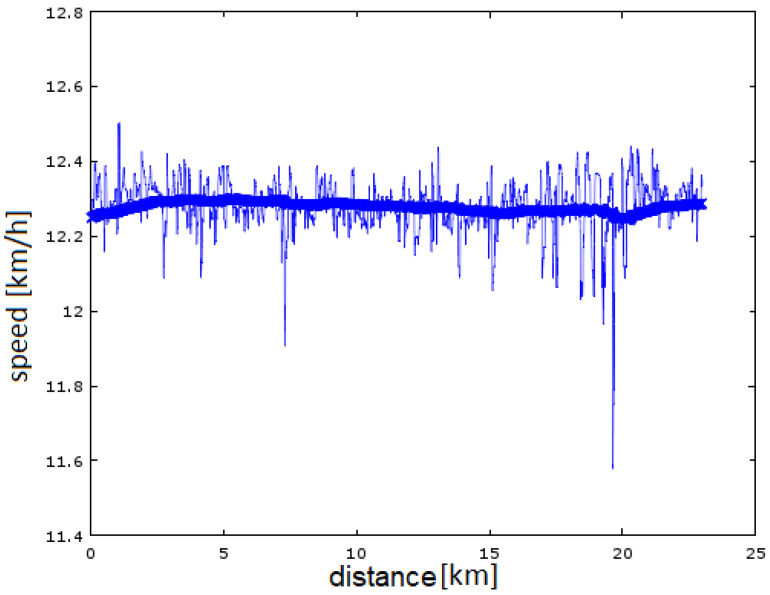
The result of velocity estimation for steady-state running (Polar watch data).

**Figure 7 sensors-23-02678-f007:**
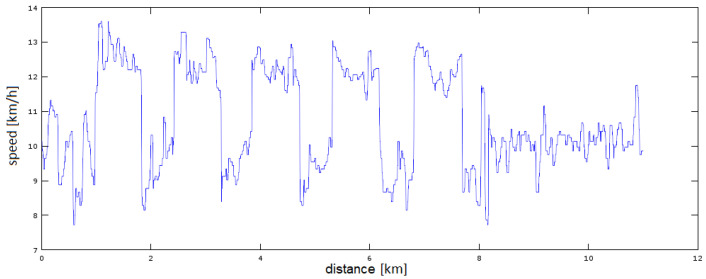
Running interval before filtration.

**Figure 8 sensors-23-02678-f008:**
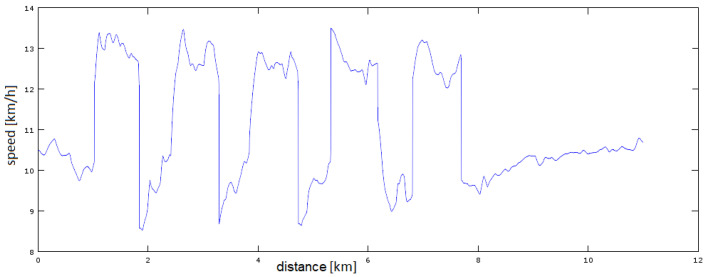
Running interval after filtration.

**Figure 9 sensors-23-02678-f009:**
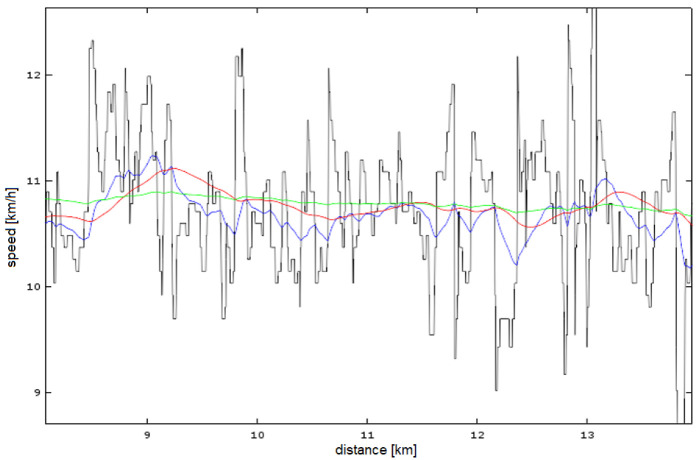
Quality of speed trend estimation: GNSS measurement (Samsung phone)—black, after the first IIR filter—blue, after the second IIR filter (coefficients set I)—red and after the second IIR filter (coefficients set II)—green.

**Figure 10 sensors-23-02678-f010:**
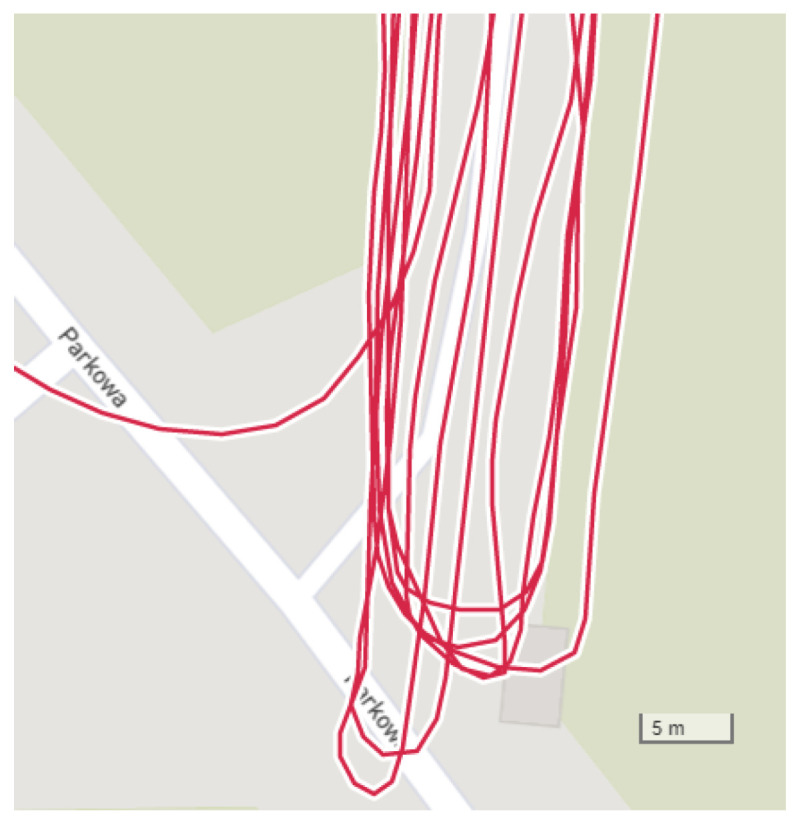
Turning around test without postprocessing (Samsung phone).

**Figure 11 sensors-23-02678-f011:**
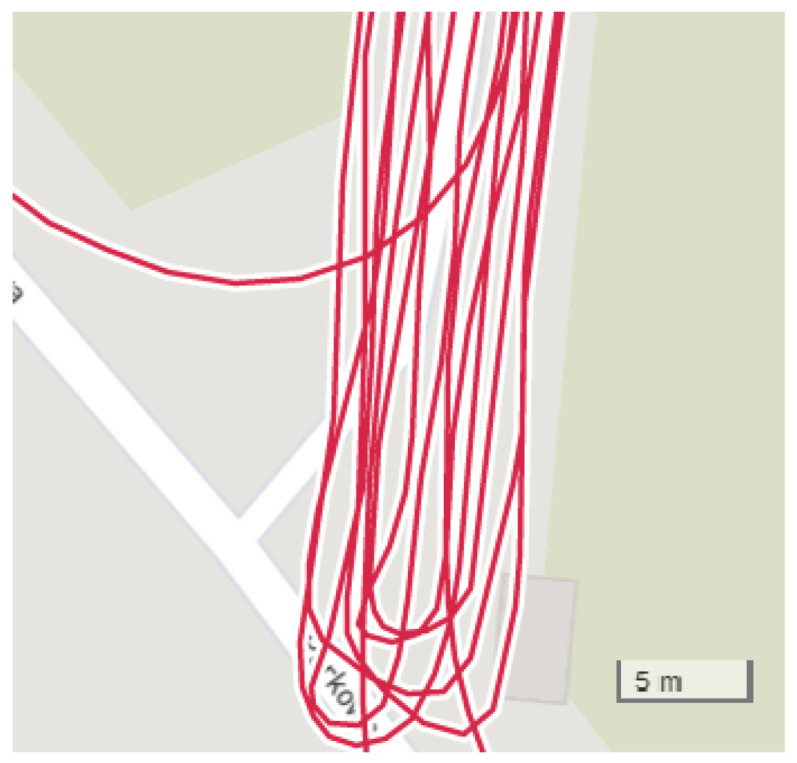
Turning around test with proposed system (Samsung phone).

**Figure 12 sensors-23-02678-f012:**
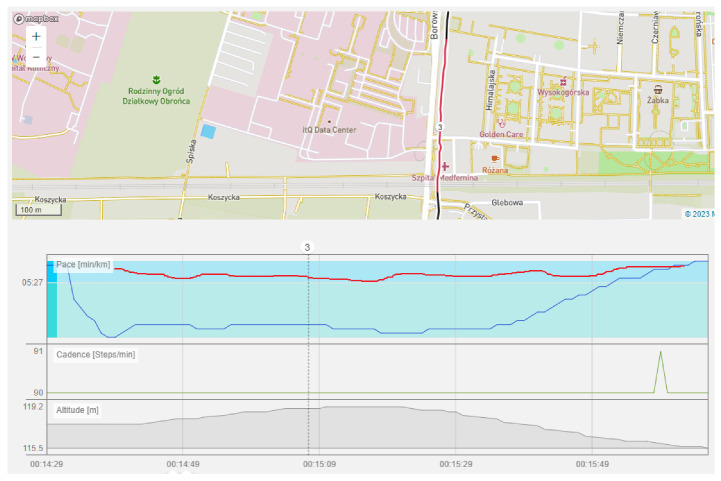
Running through the tunnel. Blue—speed without post processing, red—speed with proposed system, green—cadence (steps/min). (PolarFlow application graph).

**Figure 13 sensors-23-02678-f013:**
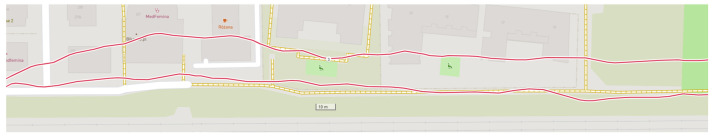
Runner route correction.

**Table 1 sensors-23-02678-t001:** Distance accuracy error.

Distance	Ordinary GNSS System (Watches)	Ordinary GNSS System (Phones)	Proposed Solution (Watches)	Proposed Solution (Phones)
5 km	0.38%	0.63%	0.19%	0.29%
10 km	0.54%	1.03%	0.33%	0.41%
20 km	0.63%	1.70%	0.39%	0.49%

**Table 2 sensors-23-02678-t002:** Speed accuracy error.

Speed	Ordinary GNSS System (Watches)	Ordinary GNSS System (Phones)	Proposed Solution (Watches)	Proposed Solution (Phones)
10 × 100 m × 100 m	0.83%	1.11%	0.53%	0.67%
10 × 200 m × 100 m	0.72%	1.03%	0.39%	0.51%
10 × 500 m × 200 m	0.68%	0.87%	0.11%	0.33%

## Data Availability

Not applicable.

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
