# Peer review of "Software Correction of Speed Measurement Determined by Phone GNSS Modules in Applications for Runners"

_sensors, 2023, doi:10.3390/s23052678_

Round 1

Reviewer 1 Report

GENERAL COMMENTS:

- ‘GPS’ is the abbreviation of Navstar GPS, not a synonym for Global Navigation Satellite Systems (GNSS). Since this abbreviation appears in almost every paragraph, the author should review the article and decide, does it refer to the USA GPS Navstar or to the global navigation satellite systems (GNSS) in general. It really makes the difference.  

- the Introduction section should contain information about similar research (application of low-pass filters to GNSS data)   

- there is ‘software correction’ phrase put in the title, but no description of it inside the article

- the values presented in tables 1 & 2 seem to be calculated on the basis of multiple (probably different shape) tracks. Also the (different) devices were grouped. Therefore the solutions (last columns) are questionable. More below in the proper comments.

- where is the Discussion section???

- numbers and units should be separated with the space (e. g. lines 22, table 1, line 179)

INTRODUCTION:

- lines 19-20: Endomondo is dead since the beginning of 2021; miCoach should be Adidas miCoach. The lack of the most popular: Strava, Runkeeper. 

- lines 21-22: ‘frequency of 1Hz’ – this is probably the recording frequency, not how often the location is calculated by the device

- line 22: ‘and from this the route, speed, acceleration are determined’ – is the Author sure, that these parameters are calculated by the GNSS’s sensors? If yes – source is needed.

- line 51: what is ‘real-life processing’?

METHODS:

- line 59: what protocol?

- line 59: how many recorded points were analyzed? There is only info about the runs.

- line 62: ‘Garmin and Polar’ – what models? Were they GPS or multi-GNSS receivers?

- line 64-65: no information about the accuracy of the reference receiver (what method was used)

- sections: 3 (lines 69+), 4 (lines 92+) and 5 (111+) should probably go to Methods section and be given names 2.1, 2.2, 2.3 consecutively;

- line 73: multipath errors should be mentioned too (in fact they play a huge role in determining the receivers’ position)

- lines 82-83 and Fig. 1 - One cannot judge what is the accuracy of the GNSS receiver by watching the route put on the map, as you do not know how accurately is the map positioned in a certain coordinate system. There were many articles published about the accuracy of GNSS-wristwatches and smartphones, among which especially those based on reliable measures (RMS, DRMS, CEP etc.) are worth mentioning.

- lines 92+, 111+: it is not clear, if the Author describes his/her own software or the one offered by the producer of a device

- Was each and every run recorded with the reference receiver attached? Were the runs conducted on the same route?  

RESULTS:

- 1st paragraph (lines 135-141) should probably go to Methods section. What more: is the Author sure, that data delivered by smartphones’ apps and sports watches is ‘raw’? Is it possible, that the data was subjected to smoothing process (or other) before downloading?

- Fig. 5, 6, 7: on the example of which receiver these data were presented?

- Fig. 6,7: distance not distanse

- line 180: pace not peace

- table 1 & 2: the size of error will be different for different speeds (and for different receivers), as well as it will depend on the shape of a route (turns, etc.). Therefore suggesting a single, uniform solution (last column) is wrong.

- table 1 & 2: columns 2 & 3 - are these results the average values of these devices? E.g. column 2 corresponds to the average error achieved by both watches?, column 3 – both smartphones?  

CONCLUSIONS:

- some of them were obvious before this research;

- what this type of research (procedure or results) can be used for? Who is it addressed to? These questions can be answered only on the basis of the corrected results tables (see last two comments to the Results’ section above)

 REFERENCES:

- the structure of this section is weird: it’s neither in alphabetic order nor the order of appearance in the text

- when writing about technology and rapid development in this area, one should avoid old publications/research

- not sure if items #5 (line 214), #13 (230), #17 (238), #18 !!! (line 240), #20 (244) should appear in the article; their value is low

- messy structure of items (please refer to the journal’s guidelines)

ABSTRACT:

- line 2: ‘mounted’ is not the proper word here

- line 4-5: there is no information about uphill running inside the article (did the Author analyze it?)

Author Response

Hello,

Thank You for Your comments and time spending on reading my article.

I made the following changes to the article:

  1. References are in alphabetic order.
  2. Distinguish between GPS and GNSS.
  3. Changed phones APPs names.
  4. For Polar and Garmin watches added GNSS systems.
  5. For reference receiver added accuracy.
  6. In the article my own software system is described.
  7. Results used in the tables are the mean values. I didn’t distinguished between Samsugn and Xiaomi. The same was for Polar and Garmin. Those equipments (phones and watches) represent the same accuracy (catalogue data).
  8. Added Discussion section.
  9. Added figure 1 with route, where the tests were done.
  10. Improved Result section by:
  • Adding results for “turning around” test (figures 10, 11)
  • Adding quality improvement with one and two IIR filters
  • Distinguish between proposed solution results for phones and watches
  1. Added few new citations.

Best Regards,

Reviewer 2 Report

check for minor typo mistakes.

Author Response

Hello,

Thank You for Your comments and time spending on reading my article. 

I made the following changes to the article:

  1. English spell checked.
  2. Added Discussion section.
  3. Improved result section.
  4. Added few new citations.

Reviewer 3 Report

Over all, this paper presents promising results and can be accepted for publication with minor revisions

1.     Suggested the authors to include latest references.

2.     The authors should check for the typos. (Ex: In occlusion, some places marked "," instead of "." at the end of the sentences)

3.     In Table 1 & 2, the parenthesis are not closed.

4.     In conclusion change the word "sport watch" to “sports watch".

5.     Suggested the authors to mention the MATLAB version which they are used to simulate the results.

6.     In Figure 4, change the term Filter 1 IIR to "IIR Filter 1" and also in second block to "IIR Filter 2"

7.     Suggested the authors to check for the typos like some of the sentences are starts with small alphabets. Kindly check.

Author Response

Hello,

Thank You for Your comments and time spending on reading my article. 

I made the following changes to the article:

  1. English spell checked.
  2. Added Discussion section.
  3. Improved result section.
  4. Added few new citations.
  5. Tables corrected.
  6. Figure 4 corrected.
  7. Matlab version added.

Best Regards,

Round 2

Reviewer 1 Report

Dear Author,

thank you very much for the extensive modifications. However I can still see some minor issues: 

- line 209: new sentence should start with "T",

- fig. 12 needs the source of the graphics/printscreen if it is not Author's own graphics,

- References, item #19 & #20 - the same article; item #24 - lack of the year. 

What more, in my first review I had some comments to the References section (messy structure, appearance of some items). Two comments are still valid:  

- items need line-by-line check. E.g. line 258: Ashby N., .... whereas in line 259: Bardus M, Smith JR, .... 

Let's cite 'Instructions for Authors': "Your references may be in any style, provided that you use the consistent formatting throughout. It is essential to include author(s) name(s), journal or book title, article or chapter title (where required), year of publication, volume and issue (where appropriate) and pagination."

- in my opinion such items as: #7, #14, #15, #23 (PhDs, not-published, conference) should not appear in any valuable article, their value is low: 

#7 - Huerne - PhD Thesis; #14 - Miller - no year, no publisher, not published yet ; #15 Miller - PhD Thesis; #23 Stathas - conference paper

sincerely,

Author Response

Hello,

Thank You for more comments:

Changes I made:

  1. removed items #7, #14, #15, #23 from References.
  2. made improvements to References (it was very messy).
  3. added reference to fig. 12.

Best Regards,